# The association of macronutrients in human milk with the growth of preterm infants

Yi-Hsuan Lin[1], Ya-Chi Hsu[1], Ming-Chih Lin[1,2,3]*, Chao-Huei Chen[2,4], Teh-Ming Wang[1]

**1** Division of Neonatology, Department of Pediatrics, Taichung Veterans General Hospital, Taichung, Taiwan, **2** School of Medicine, National Yang-Ming University, Taipei, Taiwan, **3** Department of Food and Nutrition, Providence University, Taichung, Taiwan, **4** Center for Faculty Development, Taichung Veterans General Hospital, Taichung, Taiwan

* mingclin@gmail.com

**Data Availability Statement:** All relevant data are within the manuscript and its Supporting Information files.

**Funding:** The author(s) received no specific funding for this work.

## Abstract

### Background

Breast milk is the optimal choice for feeding premature babies. However, the prevalence rate of extrauterine growth restriction in preterm infants remains high.

### Objectives

The purpose of this study was to analyze the macronutrients present in human milk and the correlation with the growth of in-hospital preterm infants.

### Methods

This prospective study is based on data from 99 in-hospital preterm infants younger than 37 weeks of gestational age on an exclusively human milk diet. Infants who had previously received parenteral nutrition were eligible, but they had to have reached full enteral feeding at the time that the samples were taken. A total of 3282 samples of raw human milk or donor pasteurized milk were collected. The levels of lactose, protein, fat, and energy in the samples were measured using a Miris human milk analyzer. The primary outcome was weight growth velocity (g/kg/day) which was obtained using two-point approach.

### Results

The mean (±standard deviation) macronutrient composition per 100 mL of milk was 7.2 (±0.3) g of lactose, 1.1 (±0.2) g of true protein, 3.5 (±0.9) g of fat, and 66.9 (±6.5) kcal of energy. The protein concentration in human milk had a positive, significant correlation with body weight gain, with a coefficient of 0.41 (p < 0.001). After adjusting for gestational age, postmenstrual age, small-for-gestational age, intraventricular hemorrhage, patent ductus arteriosus or congestive heart failure, duration of total parenteral nutrition support, bottle feeding or use of orogastric tube, and ventilator support, total daily protein intake was associated with body weight growth (p < 0.001).

**Competing interests:** The authors have declared that no competing interests exist.

## Conclusion

Both the protein concentration in human milk and the daily total protein intake had a positive correlation with the body weight gain of premature infants. Routine analysis of breast milk and individualized fortification might be indicated to optimize the growth of preterm infants.

## 1. Introduction

Breast milk has been reported to confer numerous benefits for hospitalized preterm infants, including improved feeding tolerance, shortened time to achieve full enteral feeding, lower incidence of necrotizing enterocolitis (NEC), and lower sepsis rate [1–6]. Furthermore, studies have revealed that breast milk confers long-term protective effects against neurodevelopmental disabilities, childhood obesity, and metabolic syndrome in adulthood [1, 6, 7].

Nevertheless, premature infants require high protein intake to achieve catch-up growth [8–10]. The nutrients provided by unfortified human milk are not sufficient for the growth needs of premature babies, especially certain nutrients such as protein, calcium, and phosphate [6]. Moreover, the prevalence rate of extrauterine growth restriction (EUGR) in preterm infants remains high, even when they are fed human milk with standard fortification [8, 11–14]. The risk of EUGR is highest in preterm infants fed predominantly donor human milk [12, 13]. EUGR is a key prognostic factor, especially of poor neurodevelopmental outcomes, in preterm infants [15–17]. We hypothesized that the macronutrient concentration in human milk, particularly its protein content, may be associated with the growth of preterm infants. Therefore, we analyzed the macronutrients present in human milk to determine any correlations with the body weight gain of in-hospital preterm infants.

## 2. Materials and methods

This prospective study was conducted in the neonatal intensive care unit of a tertiary center located in central Taiwan from August 2015 to April 2017. The patients included were preterm infants less than 37 weeks of gestational age who had reached full enteral feeding, did not need parenteral nutrition support at the time that the human milk samples were collected, and were on an exclusively human milk diet with their own mothers' milk or pasteurized donor milk. Infants were excluded if they died prior to hospital discharge, if they had a major congenital abnormality, or if their mothers refused to participate or had insufficient human milk samples. Insufficient human milk sample was defined as more than one missing sample within a week. Finally, total 99 infants were enrolled, as illustrated in Fig 1.

The donor human milk fed to infants in this study was obtained from the Taipei City Hospital Milk Bank, which was established in 2005. The qualified donor milk was pasteurized (62.5˚C for 30 minutes). Milk donors were unpaid volunteers, most of whom delivered term infants (91.2%). Most of the donor milk was mature milk because 96.5% of mothers donated their human milk at >1 month postpartum [18]. The protocol of this study was approved by the Institutional Review Board of Taichung Veterans General Hospital, and informed consent was obtained from all mothers.

### 2.1 Human milk analysis

In consideration of the wide variations in human milk composition (e.g., higher lipid content in hindmilk), we collected samples from the human milk that was fed to infants twice a day and calculated the average of the two analyses' results. To reduce the risk of day-to-day variations of the macronutrient level in human milk, a total of 245 weekly averages of nutrient

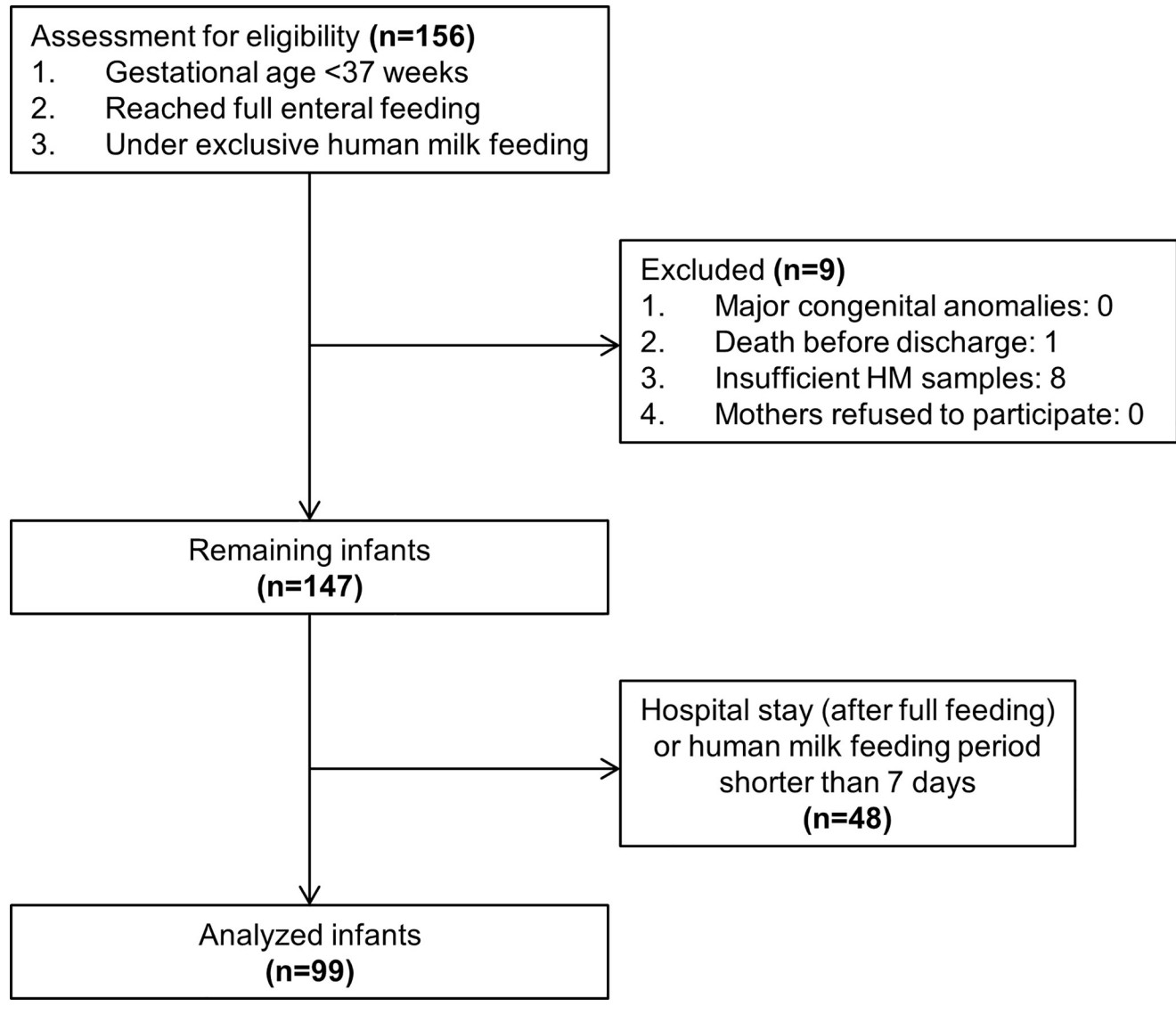

**Fig 1. Participant flow through the trial.**

contents were used for analysis. The collection details are displayed in Fig 2. Some infants contributed to multiple data points if the hospitalization period was long. The number of data points each infant contributed was reported in S1 Fig.

A total of 3282 samples of raw human milk or donor pasteurized milk were collected. These samples were stored at −20˚C. Before analysis, each sample was warmed to 37–40˚C and homogenized. A 3-mL sample was analyzed during each measurement using a Miris human milk analyzer (HMA). The Miris HMA has been proven to be a reliable and helpful tool for bedside analysis of human milk based on mid-infrared transmission spectroscopy [19–22]. A check procedure with zero level adjustment and a replicate samples control were the required quality controls for Miris HMA. A zero-setting check procedure was implemented routinely at startup and after cleaning the instrument (i.e., every 10th analysis). A monthly repeatability test was also performed. Lactose, fat, crude protein, and true protein were measured in grams (g) per 100 mL of milk, and energy values were measured in kcals per 100 mL. The crude protein category—that

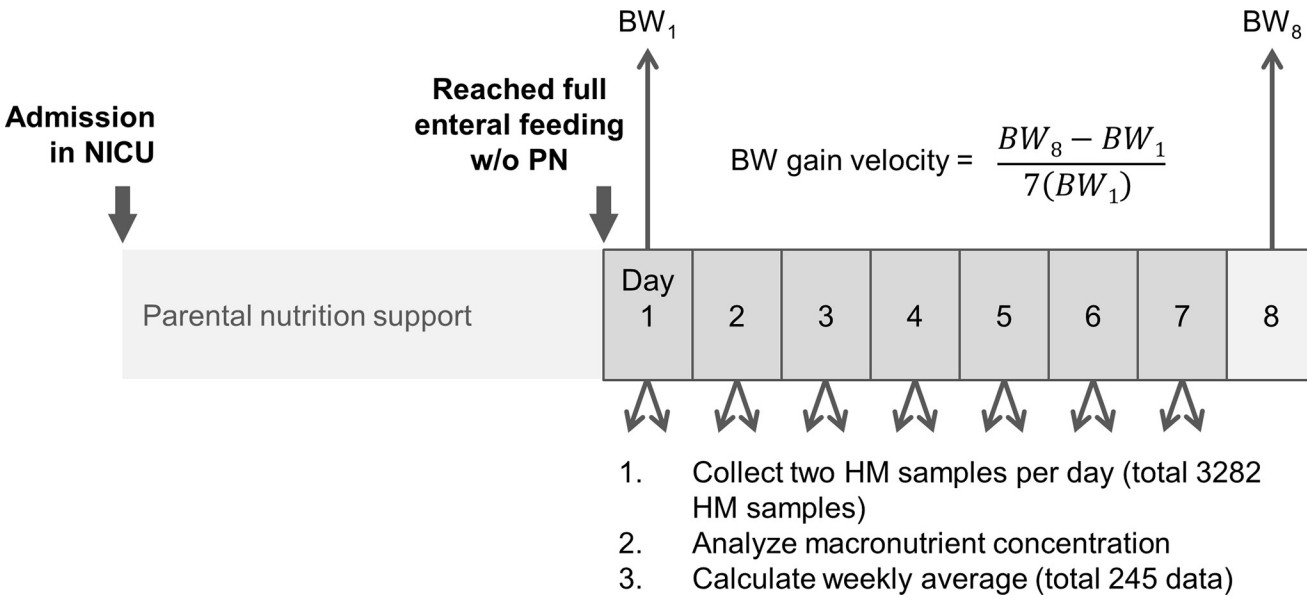

**Fig 2. Time relationship between milk analysis and growth measurement.**

is, the total nitrogen level—included all protein and nonprotein nitrogen sources. True protein covered only the protein sources of nitrogen. The fat level corresponded to the total lipid-soluble content including human milk triglycerides, diglycerides, free fatty acids, phospholipids, and cholesterol. The energy value was calculated using the following formula: total
energy = $9.25 \times$ fat + $4.40 \times$ total nitrogen + $3.95 \times$ lactose [20].

## 2.2 Enteral feeding data

Nutritional intake data were collected from nursing charts. The analyzed human milk samples were all from unfortified milk. Before the milk was fed to the infants, fortifier was added as the tolerated dose (i.e., a maximum of 4 packets per 100 mL) at the bedside. The amount of human milk fed to the infants was recorded in milliliter/kilogram/day (mL/kg/d). The brand (Similac HMF or Enfamil HMF) and dosage of human milk fortifier used were also recorded. Subsequently, included the energy and protein amount from fortifier, the total protein intake (g/kg/d) and protein-to-energy (PE) ratio (g/100 kcal) were calculated.

## 2.3 Outcome measurement

The primary outcome was growth velocity. To reduce the effect of day-to-day variations in body weight measurements, weight growth velocity (g/kg/day) was obtained by subtracting the first from the last weight of the week divided by 7[23, 24], as illustrated in Fig 2. We used the weight on the first day of the week as the kilogram denominator. The 2013 Fenton growth chart was used to report the percentiles and z scores of weight [25]. Head circumference and length growth velocity were also calculated and expressed in centimeters/week (cm/wk).

## 2.4 Cofactors

Clinical and demographic variables were obtained from medical records. We collected potential confounding factors including small-for-gestational age(SGA), intraventricular hemorrhage (IVH) more than grade 1, patent ductus arteriosus (PDA), congestive heart failure

(CHF), NEC, duration of total parenteral nutrition (TPN) support, bottle feeding or feeding through an orogastric tube, and use of ventilator support. SGA was defined as a birth body weight below the 10th percentile of the standard fetal growth curve. NEC was defined as definite or advanced disease (≥stage IIA) according to Bell's modified staging criteria. The cofactor of bottle feeding means oral feeding without an orogastric tube applied on some days of the week when samples were taken. Ventilator support was defined as nasal intermittent positive pressure ventilators or nasal continuous positive airway pressure used on some days of the week when the sample was taken. Gestational age at birth and postmenstrual age were also included as potential confounding factors. According to definitions provided by the World Health Organization, we classified all infants into two subcategories based on gestational age: extremely and very preterm (less than 32 weeks) and moderate to late preterm (32 to <37 weeks).

### 2.5 Statistical analysis

Data are presented as the mean, standard deviation (SD), and 95% confidence interval (CI). The distribution of the duration of TPN support was relatively skewed. Consequently, this variable was reported as the median and interquartile range. To test whether the data were normally distributed, the Kolmogorov–Smirnov test was used. The main exposure variables (protein concentration in human milk and total protein intake amount) and outcome variable (body weight gain velocity) were all normally distributed. Pearson's correlation coefficients were calculated to analyze the relationship of macronutrients in human milk with the growth of in-hospital preterm infants. Univariate analysis for possible confounding factors was performed. All the factors significant (p<0.05) on univariate analysis are entered in the multivariate analysis model. In addition, PDA, SGA and gestational age at birth were known important cofactors for weight growth of preterm infants [26, 27]. We also included these three factors in the multivariate analysis model, even not statistically significant in univariate analysis. Because some infants contributed more than 1 weekly sample if their hospital stay was longer, these data were not independent. We used generalized estimating equations to control for potential confounding factors. Statistical significance was defined as a p-value less than 0.05. All data analyses were performed using SPSS version 22 and SAS 9.4.

## 3. Results

The demographic data of 245 weekly samples are summarized in Table 1. The mean gestational age was 29.7 weeks, and the mean birth body weight was 1242 g. On average, patients received full enteral feeding with 150.8 mL/kg/day of human milk, comprising 114.2 kcal/kg/day. After adding human milk fortifier, the mean protein intake was 2.7 g/kg/day, and the PE ratio was 2.4 g/100 kcal. The mean body weight gain was 14.1 g/kg/day. In addition, the demographic data of 99 included preterm infants was displayed in S1 Table. The results of human milk analysis using the Miris HMA are summarized in Table 2.

The protein concentration in human milk was positively correlated with body weight gain, with a correlation coefficient of 0.41 (p < 0.001) (Table 3). The daily protein intake also had a positive association with the growth of premature infants, with a correlation coefficient of 0.47 (p < 0.001). The linear relationship between protein and rates of weight gain is presented in Fig 3.

After adjusting for confounding factors, the protein concentration in human milk and total daily protein intake were still associated with body weight growth (Table 4). Each gram per kilogram per day of additional protein was estimated to increase the velocity of weight gain by 3.63 g/kg/day (Table 5).

**Table 1. Demographic data.**

| Variables | Weekly sample (n = 245) |
|---|---|
|  | mean±SD |
| Gender |  |
| Male | 147 (60.0%) |
| Female | 98 (40.0%) |
| Small for gestational age | 49 (20.0%) |
| IVH ($\geq$ grade 2) | 12 (4.9%) |
| Significant PDA | 6 (2.4%) |
| Necrotizing enterocolitis | 38 (15.5%) |
| Donor milk use[#] | 31 (12.7%) |
| Gestational age at birth, weeks | 29.7±2.7 |
| Extremely preterm (<28 weeks) | 77 (31.4%) |
| Very preterm (28 to 32 weeks) | 99 (40.4%) |
| Moderate to late preterm (32 to 37 weeks) | 69 (28.2%) |
| Birth body weight, g | 1242.0±389.3 |
| Body weight, Z-score[*] | -1.9±1.2 |
| Body weight gain, g/kg/day | 14.1±5.6 |
| Feeding amount, ml/kg/day | 150.8±7.7 |
| Total energy intake, kcal/kg/day[**] | 114.2±11.6 |
| Total protein intake, g/kg/day[**] | 2.7±0.5 |
| Protein/energy ratio, g/100kcal[**] | 2.4±0.4 |
| Fortifier amount, packets/100ml | 2.6±0.8 |
| Duration of TPN support, day | 18 (0, 33) [&] |
| Post menstrual age, weeks[*] | 35 (34,37) [&] |

IVH: intraventricular hemorrhage; PDA: patent ductus arteriosus; TPN: total parenteral nutrition; SD: standard deviation

[#] Total or partial donor milk was fed to the infants during the week.

[*] Infant age (postmenstrual age) and weight status (z-score) at the time of growth measurements (the first day of week)

[**] The energy and protein amount of the human milk fortifier were included.

[&] The variable is reported as the median (25%ile, 75%ile).

## 4. Discussion

We demonstrated a positive linear association between the protein concentration in human milk and the weight gain of premature infants. There was no significant correlation between other human milk macronutrients and growth, and the daily energy intake was only weakly correlated with weight gain. The results revealed the importance of the protein content of milk

**Table 2. Macronutrient concentration in human milk samples (N = 245).**

|  | mean±SD | range |
|---|---|---|
| Energy, kcal/100ml | 66.9±6.5 | (51.6–88.7) |
| Lactose, g/100ml | 7.2±0.3 | (6.0–7.9) |
| True protein, g/100ml | 1.1±0.2 | (0.5–2.5) |
| Lipid, g/100ml | 3.5±0.9 | (1.9–9.7) |

SD: Standard deviation

**Table 3. Correlation with the velocity of body weight gain (g/kg/day).**

| Variables | Pearson correlation coefficient | p-value |
|---|---|---|
| Components of human milks | | |
| Energy | 0.16 | 0.01 |
| Lactose | 0.03 | 0.61 |
| Protein | 0.41 | <0.001 |
| Lipid | 0.08 | 0.20 |
| Daily energy intake | 0.17 | 0.01 |
| Daily protein intake | 0.47 | <0.001 |
| Protein/ energy ratio | 0.36 | <0.001 |

and PE ratio for the growth of in-hospital premature infants, which is consistent with the results of other studies [28–31].

The study conducted by Kashyap and Heird [28] revealed that each gram per kilogram of additional protein increased the rate of weight gain by 3.44 g/kg/day. In the studies by Olsenet al. [29] and Ernst et al. [30], a single gram per kilogram of additional protein was associated with an increase in rate of weight gain of 4.1 g/kg/day and 6.5 g/day (equivalent to 4.3 g/kg/day if weight is assumed to be 1.5 kg), respectively. The estimated increment was similar in our observational study. Our model based on a generalized estimating equation predicted that a single gram per kilogram per day of additional protein intake increased the velocity of weight gain by 3.63 g/kg/day (Table 5).

Nevertheless, these studies were conducted in the late 20th century in Western countries. Clinical practice and nutrition guidelines have changed substantially over the past two decades. Moreover, most of these studies focused on the effect of parenteral nutrition in the early growth stage of premature infants, and they assumed the human milk energy content to be 67 kcal/100 mL and protein content to be 1.2–1.4 g/100 mL. A strength of our study is that

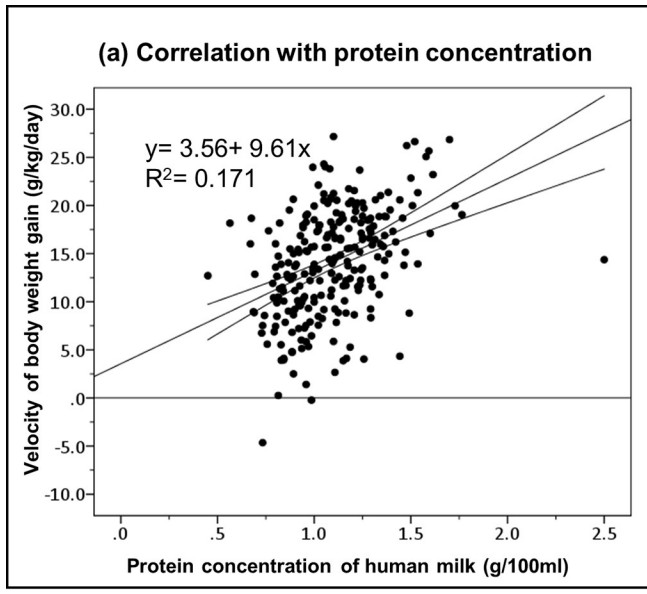
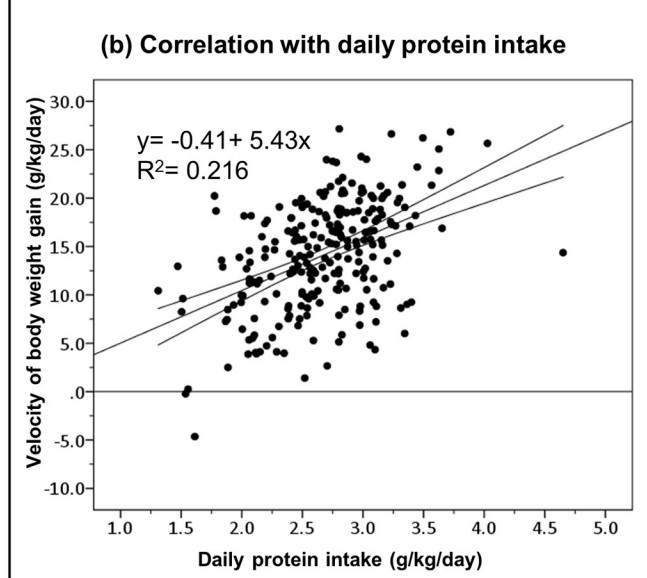

**Fig 3. Linear relationship between protein content and body weight gain.** The correlation of the velocity of body weight gain with (a) protein concentration in human milk and (b) daily protein intake. Spots in the figure were unadjusted values. The simple linear regression equation is indicated by the straight line. The upper and lower curves represent the 95% confidence interval of the mean.

**Table 4. Relationship between protein concentration in human milk and body weight gain: Results from a generalized estimating equation.**

| Variables | b | lower limit of 95% CI | upper limit of 95% CI | p-value |
|---|---|---|---|---|
| Protein concentration of HM | 3.66 | 0.06 | 7.26 | 0.046 |
| Bottle feeding* | -2.41 | -4.15 | -0.66 | <0.01 |
| Ventilator support** | 0.15 | -1.49 | 1.78 | 0.86 |
| Small for gestational age | 1.77 | -0.54 | 4.09 | 0.13 |
| Intraventricular hemorrhage | 0.86 | -0.90 | 2.62 | 0.34 |
| PDA or CHF | -1.71 | -4.68 | 1.26 | 0.26 |
| Duration of TPN | -0.08 | -0.13 | -0.03 | <0.01 |
| Gestational age at birth | -0.16 | -1.86 | 1.54 | 0.85 |
| Post menstrual age | -0.55 | -0.96 | -0.15 | <0.01 |

b: unstandardized coefficients; CI: confident interval; HM: human milk; PDA: patent ductus arteriosus; CHF: congestive heart failure; TPN: total parenteral nutrition

* Bottle feeding was applied on some days of the week when samples were taken.

** Noninvasive positive pressure ventilators or nasal continuous positive airway pressure was used on some days of the week when the sample was taken.

we enrolled preterm infants who had reached full enteral feeding with exclusively human milk in an East Asian population. We analyzed human milk samples for the actual level of macronutrients every day, because the composition of breast milk is dynamic [28–30].

Several studies have been conducted to determine the protein requirement and optimal PE ratio in preterm babies. Ziegler [9] and ESPGHAN [32] recommended an enteral protein intake (PE ratio) of 4.0–4.5 g/kg/day (3.6–4.1 g/100 kcal) in premature infants with a body weight of <1200 g and 3.5–4.0 g/kg/day (2.8–3.6 g/100 kcal) in those with a body weight of >1200 g. That is, a protein concentration of approximately 2.8 g per 100 mL of human milk must be achieved. However, the present study revealed that only 2.7 ± 0.5 g/kg/day protein intake and a mean PE ratio of 2.4 ± 0.4 g/100 kcal were achieved, even with standard fortification (Table 1). In our clinical practice, human milk fortifier was initiated when enteral feeding reached 100 mL/kg/day, and the dosage was titrated to a maximum of 4 packs of powder fortifier (1.0–1.1 g of protein content) in 100 mL of human milk. Other studies have also revealed

**Table 5. Relationship between daily protein intake and body weight gain: Results from a generalized estimating equation.**

| Variables | b | lower limit of 95% CI | upper limit of 95% CI | p-value |
|---|---|---|---|---|
| Daily protein intake# | 3.63 | 2.36 | 4.90 | <0.001 |
| Bottle feeding* | -2.05 | -3.69 | -0.41 | 0.01 |
| Ventilator support** | 0.10 | -1.40 | 1.61 | 0.89 |
| Small for gestational age | -1.32 | -0.87 | 3.51 | 0.24 |
| Intraventricular hemorrhage | 0.97 | -0.50 | 2.44 | 0.20 |
| PDA or CHF | -0.85 | -3.97 | 2.26 | 0.59 |
| Duration of TPN | -0.07 | -0.11 | -0.02 | <0.01 |
| Gestational age at birth | -0.38 | -1.96 | 1.21 | 0.64 |
| Post menstrual age | -0.57 | -0.92 | -0.22 | <0.01 |

b: unstandardized coefficients; β: standardized coefficients; HM: human milk; PDA: patent ductus arteriosus; CHF: congestive heart failure; TPN: total parenteral nutrition

# The energy and protein amount of human milk fortifier were included.

* Bottle feeding was applied on some days of the week when samples were taken.

** Noninvasive positive pressure ventilators or nasal continuous positive airway pressure was used on some days of the week when the sample was taken.

that the current standardized fortification of human milk is often insufficient for meeting protein needs [9, 33].

One of the main causes of inadequate protein intake even after standard fortification is the considerable variations of protein components in human milk. The composition of human milk varies within feeds, with the lactation period, and differs between the mothers of preterm or term infants [34]. In our study, a human milk analysis revealed wide variation in human milk macronutrients (Table 2). In particular, the protein content of banked donor milk was lower (0.9–1.2 g/dL) than that of expressed maternal milk [34–37] because it was usually donated by women who had delivered a term infant. Therefore, the components were similar to those of mature milk. A fixed-dose fortification may result in unstable nutrient contents and potential growth failure in premature infants, and the risk is highest in those fed banked donor milk [12, 35, 38–40]. To address this problem, periodic analysis using an HMA may provide useful information and guide individualized fortification. Numerous studies have also demonstrated the feasibility and effectiveness of targeted fortification, which involves measuring the protein content of human milk and adding protein accordingly to achieve the target level recommended by nutritional guidelines [11, 35, 39, 41–46].

This observational study had some limitations. First, we measured the head circumference and body length weekly and calculated the growth velocity in centimeters/week (cm/wk). However, some of the growth velocities obtained were negative. A possible explanation is that the time interval was too short to represent the subtle change in head circumference and length. Because of inaccurate measurement, we could not analyze the relationship among head circumference, length growth, and daily protein intake. Additionally, the participants were not stratified by birth body weight considering the small sample size, but the recommended protein requirement generally varies with weight. Extremely low-birth-weight infants might particularly benefit from a higher protein intake. Finally, we only enrolled in-hospital premature infants with a gestational age of 25 to 36 weeks who reached full enteral feeding and under exclusive human milk diet. Whether the protein content in human milk also plays a crucial role in the premature infants under mixed feeding with formula milk, with parental nutrition support, and the infants after hospital discharge, remains to be determined.

A strength of our study is that we determined the actual level of human milk macronutrients based on daily milk analyses, and the detailed feeding amount including fortifier dosage was recorded to accurately calculate the total daily protein and energy intake. In addition, we only enrolled preterm infants who had reached full enteral feeding with human milk to eliminate the effect of parenteral nutrition or infant formula. Furthermore, recent data on human milk macronutrients and the growth of preterm infants in the population of Asia were lacking. We conducted a domestic and modern study on the macronutrients of human milk and their correlation with the growth of in-hospital premature infants in Taiwan.

## 5. Conclusion

Both the protein concentration in human milk and daily total protein intake have a positive association with the body weight gain of premature infants. Routine analysis of breast milk and individualized fortification may be indicated to optimize the growth of preterm infants.

## Supporting information

**S1 File. Original data.**
(XLSX)

**S1 Fig. The number of data points each infant contributed.**
(TIF)

**S1 Table. Demographic data for included infants.**
(DOCX)

## Acknowledgments

This manuscript was edited by Wallace Academic Editing.

## Author Contributions

**Conceptualization:** Ya-Chi Hsu, Ming-Chih Lin, Chao-Huei Chen, Teh-Ming Wang.

**Data curation:** Yi-Hsuan Lin, Ya-Chi Hsu.

**Formal analysis:** Yi-Hsuan Lin, Ming-Chih Lin.

**Methodology:** Yi-Hsuan Lin, Ya-Chi Hsu, Ming-Chih Lin.

**Supervision:** Ming-Chih Lin.

**Writing – Original Draft:** Yi-Hsuan Lin.

**Writing – review & editing:** Ming-Chih Lin.

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
