## [Decision Letter · Decision Letter 0]

24 Sep 2019

PONE-D-19-17439

The impact of macronutrients in human milk on the growth of preterm infants

PLOS ONE

Dear Dr. Lin,

Thank you for submitting your manuscript to PLOS ONE. After careful consideration, we feel that it has merit but does not fully meet PLOS ONE’s publication criteria as it currently stands. Therefore, we invite you to submit a revised version of the manuscript that addresses the points raised during the review process.

Major concerns were raised by reviewers with regards to insufficient information provided in methods, results presentation and implications of research. A clear case for the rationale and novel contributions to literature is required. Manuscript would also benefit from careful review for language and writing throughout. 

We would appreciate receiving your revised manuscript by Nov 08 2019 11:59PM. To enhance the reproducibility of your results, we recommend that if applicable you deposit your laboratory protocols in protocols.io, where a protocol can be assigned its own identifier (DOI) such that it can be cited independently in the future. For instructions see: http://journals.plos.org/plosone/s/submission-guidelines#loc-laboratory-protocols

We look forward to receiving your revised manuscript.

Kind regards,

Melissa F. Young, Ph.D.

Academic Editor

PLOS ONE

Journal Requirements:

Additional Editor Comments (if provided):

Reviewers' comments:

Reviewer's Responses to Questions

**Comments to the Author**

1. Is the manuscript technically sound, and do the data support the conclusions?

Reviewer #1: No

Reviewer #2: Yes

Reviewer #3: Yes

2. Has the statistical analysis been performed appropriately and rigorously? 

Reviewer #1: No

Reviewer #2: Yes

Reviewer #3: No

3. Have the authors made all data underlying the findings in their manuscript fully available?

Reviewer #1: No

Reviewer #2: Yes

Reviewer #3: Yes

4. Is the manuscript presented in an intelligible fashion and written in standard English?

Reviewer #1: No

Reviewer #2: No

Reviewer #3: Yes

5. Review Comments to the Author

Reviewer #1: The manuscript under review reports macronutrient concentrations in human milk from different sources and their impact on weight gain of pre-mature infants. While topic of human milk composition in relation to pre-term infants is one that needs more exploration, the presented data is too little to justify a stand-a-lone publication.

The only components analyzed are protein, fat, and carbohydrates in human milk with a commercial analyzer, apparently without quality controls. The composition of human milk was in-fact not analyzed in this study as claimed by the authors. It is not clearly described what milk was analyzed other than human milk. What type of donor milk was used? What is the difference between human and maternal milk as all the milk is human milk? Infants were included only when exclusively fed with human milk of sorts without parental nutrition support, which has not been defined and it is up for the reader to guess. Yet, all over the manuscript it is indicated that there was parental nutrition support. The sample collection details are missing, and no ex- or inclusion criteria are provided for the mothers.

Statistical analysis was adjusted for numerous factors which are not explained to why they were chosen nor how they were choses or tested to be included. Yet, infant age was not one of them. We don’t know whether the data was normally distributed, while the used tests in the paper are based on normally distributed variables.

The reader is not informed about how long milk samples were collected (lactation stage/infant age). The authors emphasize the daily protein intake as being correlated to infant weight gain, which is only a derivative of the protein in the milk, which was found to correlate in the first place. None of the found results are novel or unexpected.

The figure has no title and no explanation is given, e.g. what are the 3 lines in each of the graphs? Table 3 is not useful for the reader; the data of the significant correlation is much better displayed in a graph. The objective of your research is not a strength of this study.

These are just some of the concerns I have for this manuscript, besides the needed language revisions and corrections, e.g. p7L100: you are not sampling human milk fed infants, you are collecting milk that is fed to infants.

Reviewer #2: The manuscript describes the nutritional intake of preterm infants in Taiwan. As the authors correctly conclude, the addition of powder fortifier is insufficient to meet the premature infants' needs for nutrients

1. Should give age and weight status (z-score) of infants at time of growth measurements to delineate the degree of EUGR

2. Line 106-107: Must state length of period for which weight gain is given. "Weekly average" is too vague.

3. Need to give time relationship between milk analysis and growth measurement.

4. Need to point out that fortified and unfortified milk was analyzed

5. Lines 100-103: Milk analysis: of two measurements in a given day, the average was used. But then how were "weekly averages" calculated?

6. Lines 145-146: Need not repeat in text what the table says

7. Line 231: What is "HMA"?

8. Discussion is too long. Quote only articles that are directly related to the manuscript

9. "Breastfeeding, Breast Feeding and Maternal Nutrition" should be deleted from key words

Reviewer #3: The authors address the topic of the extent to which variability in human milk macronutrient content affects growth among preterm infants. To do this, they used mid-infrared spectroscopy to quantify the macronutrient content of human milk being ingested by preterm infants and assessed associations with concurrent weight gain. Their main finding was a positive correlation between the protein content of human milk (or total protein intake) and weight gain. Overall the paper's objective, design, and results are clearly described and the methods chosen are generally appropriate. The results shown support the author's conclusions. Nevertheless, there are several opportunities for improvement; in particular, the authors could clarify the methods in certain sections (described below), clarify the method of statistical analysis, consider additional covariates, and improve the organization and readability of the Discussion section. Further discussion of each of these suggestions is provided below.

MAJOR SUGGESTIONS:

The methods section requires additional details and the analysis plan requires clarification:

1. Please report the total number of infants included in the study, and the number of measurements taken per infant.

2. From the authors' description it seems that each baby could contribute multiple data points (e.g. if a baby was in the hospital for 5 weeks it could contribute 5 different data points to the study, an average milk content and average growth rate for every week). If multiple data points from the same baby were used in the analysis, then the data aren't independent and the statistical analysis needs to account for that (for example, using generalized estimating equations to account for correlated data due to repeated measurements on the same infant). Alternatively, the authors could consider using the average protein intake and growth rate for each baby over the entire study period so that each infant contributes just one data point, and adjusting for length of stay as a covariate.

3. The authors should investigate adding gestational age at birth as a covariate. Gestational age is associated with protein content in maternal milk and with growth, and could act as either a confounder, a mediator, or possibly an effect modifier.

MINOR SUGGESTIONS:

OVERALL: The authors refer to "catch-up growth" and "catch-up growth stage" throughout the paper. Can they clarify what they mean by this (crossing percentiles upward on the growth chart? rapid growth? a critical period for growth?) and perhaps use a more specific term?

ABSTRACT:

- In the methods section, I suggest rewording the first sentence to clarify that infants who received PN previously were eligible; the criterion was to not be receiving PN at the time the sample was taken.

- I also recommend reporting the number of infants in the study in addition to the total number of milk samples, and adding the effect size of the association between protein intake and growth in addition to the p-value.

- In the conclusion, the statement "protein intakes have positive impact on the body weight gain" sounds causative but this study only shows associations; I suggest rewording this statement.

INTRODUCTION: This is a clear and concise summary of the relevant background literature and nicely describes the research question.

- I would suggest adding a hypothesis.

- There are a few minor grammatical points: Line 71, change "prognosis" to "prognostic". Line 72, add "of" between "especially" and "poor neurodevelopmental outcomes."

METHODS:

- Can the authors comment on what proportion of eligible infants were included in the study, and if any were excluded what the reasons for exclusion were?

- The authors should describe how they chose the two-point approach to measuring growth velocity from among the alternative methods available (eg Patel et al, Calculating Postnatal Growth Velocity in Very Low Birth Weight (VLBW) Premature Infants, J Perinatal 2009).

- Please describe how extreme values were handled. In my experience, there are often a few extreme values in any relatively large dataset of milk samples analyzed by MIRIS (often due to the sample not being fully homogenized). Were there any extreme values and if so were they included or discarded? If discarded, what criteria were used to determine which samples were excluded?

- Describe what is meant by the covariate "bottle feeding" - is that bottle feeding on the day the sample was taken? Bottle feeding ever?

- Similarly, describe what is meant by "ventilator support" - on the day of the sample? Ever?

- The authors state in the discussion that they could not use length or head circumference (HC) measurements due to measurement error. It may be worth adding to the methods section that the intention was to assess length and HC, and how they determined that the measurements were too inaccurate to use.

RESULTS:

- In Table 1: The title of the right column is "weekly sample" - so are these results describing individual infants, or total samples (eg were 60% of infants in the study male or 60% of samples came from male infants?). Also clarify what is meant by "Post menstrual age;" is this the PMA at the time the sample was taken? For duration of TPN support, I suspect the variable is quite skewed; consider reporting median rather than mean and standard deviation.

- Can the authors comment on the proportion of samples that were donor milk versus maternal milk? There are well described differences between the macronutrient content of donor vs maternal milk, as the authors have described in the discussion.

- Line 52, remove "Nevertheless"

DISCUSSION: Overall the organization of the discussion could be improved, and the authors could more clearly state how their findings add to what is already known on this topic. Also, while the remainder of the paper has excellent English language, the discussion has a few minor English errors that should be corrected; I have attempted to note them here to assist the authors in clarifying their language.

- The first paragraph is very long. Consider breaking it up into several paragraphs and clearly identify the key point of each paragraph.

- Line 187, add that the linear association was positive

- Line 190, clarify what "it" refers to

- Line 191, consider changing "amount" to "content of milk" or "intake" depending on the authors' intention

- Lines 211-212: clarify what "higher" means (higher than what?)

- Line 222, consider describing the protein content of the HMF rather than the number of packs

- Line 227, clarify if the authors are referring to the current study or to previous studies

- Lines 227-235; this paragraph would benefit from better organization to determine the authors' key point

- Line 237, I think the authors mean "protein intake" rather than "fortification" here

- Line 257, by "was not provided in our results" do the authors mean it was not found in their analysis or was not studied?

- Correct the English language: Line 189, remove "the" before growth. Line 198, "observation" should be "observational." Lines 201-202 I would suggest "Clinical practice and nutrition guidelines have changed substantially over the past two decades." Line 207 change "the human milk sample" to "human milk samples." Lines 211-212, consider changing "approximate" to "support," and add "the" before "intrauterine fetus." Line 217, add "with" before "body weight." Line 221, change "while" to "when." Line 240, "systemic" should be "systematic." Line 251, "target" should be "targeted." Line 254, "consist" should be "exist." Line 260, "group" should be "groups." Lines 263-264 need to be rewritten and clarified. Line 277, "prematurity" should be "premature infants"

- Can the authors comment on generalizability of their results based on the inclusion criteria of their study?

6. PLOS authors have the option to publish the peer review history of their article (what does this mean?). If published, this will include your full peer review and any attached files.

Reviewer #1: No

Reviewer #2: No

Reviewer #3: No

---

## [Author Response · Author response to Decision Letter 0]

8 Nov 2019

Dear Dr. Young: 

Thank you for inviting us to submit a revised draft of our manuscript entitled, “The impact of macronutrients in human milk on the growth of preterm infants” to PLUS ONE. We also appreciate the time and effort you and each of the reviewers have dedicated to providing insightful feedback on ways to strengthen our paper. Thus, it is with great pleasure that we resubmit our article for further consideration. We have incorporated changes that reflect the detailed suggestions you have graciously provided. We also hope that our edits and the responses we provide below satisfactorily address all the issues and concerns you and the reviewers have noted.

To facilitate your review of our revisions, the following is a point-by-point response to the questions and comments.

To Reviewer #1: 

Thank you for your comments and suggestions. The questions are answered below: 

1. RE: The only components analyzed are protein, fat, and carbohydrates in human milk with a commercial analyzer, apparently without quality controls. The composition of human milk was in-fact not analyzed in this study as claimed by the authors. 

Ans: We agree with you. There are many important compositions of human milk, and we only could analyze the macronutrient levels. I’m sorry to mislead about that. I have removed the word “composition” and replaced it with “macronutrients” in some paragraph (Line 33, 77,240). MIRIS human milk analyzer was proved a reliable and helpful tool for bedside analysis of human milk [18-21]. We also added explanation about quality control in the methods section as follows: “The check procedure with zero level adjustment and the replicate samples control was required quality controls of the MIRIS HMA. A zero-setting check procedure was implemented routinely at start-up and after cleaning the instrument, i.e. every 10th analysis. A monthly control of repeatability test was also performed.” (Page 9, Line 117-121)

2. RE: It is not clearly described what milk was analyzed other than human milk. What type of donor milk was used? What is the difference between human and maternal milk as all the milk is human milk? 

Ans: I’m sorry about the unclear description. All milk analyzed in this study was human milk. There were two source of human milk, milk of infants’ own mothers and pasteurized donor milk. We added some explanation about the donor milk as follows: “All donor human milk fed to infants in this study was obtained from the Taipei City Hospital Milk Bank (TCHMB) which was established in 2005. The qualified donor milk was pasteurized (62.5°C for 30 minutes). Milk donors were unpaid volunteers, most of whom delivered term infants (91.2%). Most donor milk is mature milk because up to 96.5% mother donated their human milk while postpartum >1 month (96.5%).[18]” (Page 7-8, Line 93-98)

3. RE: Infants were included only when exclusively fed with human milk of sorts without parental nutrition support, which has not been defined and it is up for the reader to guess. Yet, all over the manuscript it is indicated that there was parental nutrition support. 

Ans: I had reworded the sentence in the abstract to clarify that infants who received parental nutrition previously were eligible; the inclusion criterion was not be receiving parental nutrition at the time the sample was taken. (Page 3, Line 36-39)

4. RE: The sample collection details are missing, and no ex- or inclusion criteria are provided for the mothers.

Ans: We added a new figure (Figure 1) to illustrate the inclusion /exclusion criteria and the participants flow through the trial. (Section 2. Materials and Methods, Page 7, Line 82-91). The sample collection details were also added in Figure 2. Time relationship between milk analysis and growth measurement.

5. RE: Statistical analysis was adjusted for numerous factors which are not explained to why they were chosen nor how they were choses or tested to be included. Yet, infant age was not one of them. 

Ans: We redid the statistical analysis and included gestational age and post menstrual age (PMA) as cofactors (section 2.4 cofactors, Page 11, Line 155-159). The results were showed in Table 4 and Table 5.

6. RE: We don’t know whether the data was normally distributed, while the used tests in the paper are based on normally distributed variables.

Kolmogorov–Smirnov test

Variables D P value (Pr>D)

Protein concentration (g/100ml) 0.052 0.098

Total protein intake amount (g/kg/day) 0.050 0.134

Body weight gain velocity (g/kg/day) 0.045 >0.150

Ans: We added the description about normally distribution in the section 2.5 Statistical analysis as follows: “In order to test whether the data are normally distributed, the Kolmogorov–Smirnov test was used. The variables included protein concentration of human milk, total protein intake amount, and body weight gain velocity were all normally distributed.” (Page 11, Line 164-167) The Kolmogorov–Smirnov test results are displayed as follows:

7. RE: The reader is not informed about how long milk samples were collected (lactation stage/infant age). 

Ans: We added new figure 2 to illustrate about time relationship between milk analysis and growth measurement (Page 8, Line 106-110). The gestational age at birth and infant age (PMA) at the time of milk samples collection were reported in table 1 (page 13-14, Line 186). Because we only enrolled the preterm infants who reached full enteral feeding, most human milk samples analyzed in the study were mature milk.

8. RE: The authors emphasize the daily protein intake as being correlated to infant weight gain, which is only a derivative of the protein in the milk, which was found to correlate in the first place. None of the found results are novel or unexpected.

Ans: We agree with your assessment. The crucial importance of the protein amount for the growth of in-hospital premature babies has been established. However, most studies were conducted in the late 20th century in Western countries. The clinical practice and nutrition guideline have changed much over time. Otherwise, most of these previous studies focused on the impact of parenteral nutrition in the early growth stage of premature infants, and they assumed the human milk energy content of 67 kcal/100 ml and protein content of 1.2-1.4g/100ml. The strength of our study is that we determined the actual level of human milk macronutrients based on daily milk analyses, and the detailed feeding amount including fortifier dosage was recorded to accurately calculate the total daily protein and energy intake. In addition, we only enrolled preterm infants who had reached full enteral feeding with human milk to eliminate the effect of parenteral nutrition or infant formula. We conducted a domestic and modern study in an East Asian population. (Page 23-24, Line 308-316)

9. RE: The figure has no title and no explanation is given, e.g. what are the 3 lines in each of the graphs? 

Ans: The figure was relabeled as figure 3. The title of figure was displayed in the manuscript, and we added figure legend below it (Figure 3, Page 17, Line 207-211).

10. RE: Table 3 is not useful for the reader; the data of the significant correlation is much better displayed in a graph. 

Ans: The significant correlation of protein concentration of HM and weight gain was displayed in figure 1. There was no significant correlation between other human milk macronutrients and the growth, so these data were not shown in the graph.

11. RE: The objective of your research is not a strength of this study.

Ans: Thank you for your important suggestion. We reworded the first sentence of objective to clarify that we only analyzed the macronutrients of human milk and the correlations with the growth of in-hospital preterm infants. (Page 3, Line 33)

12. RE: These are just some of the concerns I have for this manuscript, besides the needed language revisions and corrections, e.g. p7L100: you are not sampling human milk fed infants, you are collecting milk that is fed to infants.

Ans: Thank you for your comments. I had revised this sentence as your suggestion (Page 8, Line 103-104). Language revisions and corrections of all over manuscript were also done (section Acknowledgement, line 324-325)

 

To Reviewer #2: 

Thank you for your comments and suggestions. The questions are answered below: 

1. RE: Should give age and weight status (z-score) of infants at time of growth measurements to delineate the degree of EUGR

Ans: Infant age (post menstrual age) and weight status (z-score) at time of growth measurements were added in Table 1. We also used the change in weight z-score within 1 week to evaluate the weight gain velocity as follows (figure c and d). But the correlation between weight z core change and protein intake is weak, probably because the difference in z score was subtle within a short period (one week). So the results was not shown in figure 1.

2. RE: Line 106-107: Must state length of period for which weight gain is given. "Weekly average" is too vague.

Ans: We have included a new Figure 2 (Page 8, Line 110) to further illustrate the time relationship between milk analysis and growth measurement.

3. RE: Need to give time relationship between milk analysis and growth measurement. 

Ans: We have added new Figure 2 (Page 8, Line 110) to further illustrate it. Please see RE#2 above.

4. RE: Need to point out that fortified and unfortified milk was analyzed

Ans: All analyzed human milk is unfortified milk. Before fed to the infants, Human milk fortifier was added as tolerated with a maximum of 4 packets per 100ml in the bedside. The brand and dosage of human milk fortifier (Similac HMF or Enfamil HMF) were also recorded, and then total protein amount (g/kg/d) and protein-energy ratio (g/100kcal) were calculated. We have supplemented the section 2.2 Enteral Feeding data with explanations of human milk fortifier (Page 9-10, Line 130-133)

5. RE: Lines 100-103: Milk analysis: of two measurements in a given day, the average was used. But then how were "weekly averages" calculated?

Ans: We have added new Figure 2 (Page 8, Line 110) to further illustrate it. Please see RE#2 above. In consideration of the wide variation of human milk composition, ie. more lipid content in hindmilk, we collected the human milk that was fed to infants twice a day (Page 8, Line 102-106). Then we used the weekly average data to reduce the effect of day-to-day variance of body weight measurements. (Page 10, Line 139-141).

6. RE: Lines 145-146: Need not repeat in text what the table says

Ans: Thank you for your suggestion. We have deleted these sentences about Table 2. 

7. RE: Line 231: What is "HMA"?

Ans: I’m sorry about the unclear description. I had corrected the word with “MIRIS HMA” in the manucript. MIRIS HMA is an abbreviation of MIRIS human milk analyzer, which explained in page 9 line 115.

8. RE: Discussion is too long. Quote only articles that are directly related to the manuscript

Ans: Thank you for your comment. We have removed some references and compressed the discussion.

9. RE: "Breastfeeding, Breast Feeding and Maternal Nutrition" should be deleted from key words

Ans: We have removed these key words.

 

To Reviewer #3 

Thank you for your comments and suggestions. We have incorporated these suggestions throughout our paper. The details are answered below: 

MAJOR SUGGESTIONS:

1. RE: Please report the total number of infants included in the study, and the number of measurements taken per infant.

Ans: Total 99 infants was included in the study (Page 7, Line 88-89). Each infant may contribute multiple data points if long hospitalization period. Total 245 weekly data were collected (Page 8, Line 105-106). The number of measurements taken per infant was displayed as follow:

We also added new Figure 1 and Figure 2 to illustrate the participants flow and collection sample details.

2. From the authors' description it seems that each baby could contribute multiple data points (e.g. if a baby was in the hospital for 5 weeks it could contribute 5 different data points to the study, an average milk content and average growth rate for every week). If multiple data points from the same baby were used in the analysis, then the data aren't independent and the statistical analysis needs to account for that (for example, using generalized estimating equations to account for correlated data due to repeated measurements on the same infant). Alternatively, the authors could consider using the average protein intake and growth rate for each baby over the entire study period so that each infant contributes just one data point and adjusting for length of stay as a covariate.

Ans: You have raised an important question. In fact, some infants contributed multiple data points in this study as the description RE#1 above. We used generalized estimating equations to account for these correlated data. The results were displayed in Table 4 and table 5. We also supplemented the 2.5 statistical analysis section with explanations of generalized estimating equations (Page 12, lines 169-171).

3. RE: The authors should investigate adding gestational age at birth as a covariate. Gestational age is associated with protein content in maternal milk and with growth, and could act as either a confounder, a mediator, or possibly an effect modifier.

Ans: We agree with you. Gestational age at birth is an important covariate. We redid the statistical analysis and included gestational age and post menstrual age (PMA) as cofactors (section 2.4 cofactors, Page 11, Line 155-159). The results were displayed in Table 4 and Table 5. 

MINOR SUGGESTIONS:

OVERALL: 

1. RE: The authors refer to "catch-up growth" and "catch-up growth stage" throughout the paper. Can they clarify what they mean by this (crossing percentiles upward on the growth chart? rapid growth? a critical period for growth?) and perhaps use a more specific term?

Ans: We agree with you. There is no unified definition of catch-up growth. Catch-up growth is usually defined as reaching an SD score of 1 –2 SDS of the reference population, but in some studies a change >0.67 SD has been used as cut-off. In our paper, I thought catch-up growth stage is a rapid growth period after initial postnatal growth failure. We have tried to avoid this unspecific term in our paper.

ABSTRACT:

1. RE: In the methods section, I suggest rewording the first sentence to clarify that infants who received PN previously were eligible; the criterion was to not be receiving PN at the time the sample was taken.

Ans: Thank you for your suggestion. I have modified the sentence (Page 3, Line 36-39)

2. RE: I also recommend reporting the number of infants in the study in addition to the total number of milk samples, and adding the effect size of the association between protein intake and growth in addition to the p-value.

Ans: Total 99 infants was enrolled in the study, which was displayed in abstract (Line 35), section 2 Materials and Methods (Line 89) and figure 1. We set a power of 90% (description in section 2.5 Statistical analysis, line 172), and the effect sizes were 5.70 and 2.28 (reported in section 3 Results, Line 199 and 202).

3. RE: In the conclusion, the statement “protein intakes have positive impact on the body weight gain” sounds causative but this study only shows associations; I suggest rewording this statement.

Ans: We have revised the conclusion as follows: “Both the protein concentration of human milk and the daily total protein intakes have positive correlations with the body weight gain of prematurity.” (Page 4, Line 51-53)

INTRODUCTION: This is a clear and concise summary of the relevant background literature and nicely describes the research question.

1. RE: I would suggest adding a hypothesis.

Ans: We have added our hypothesis and revised the last part of introduction (Line 75-78).

2. RE: There are a few minor grammatical points: Line 71, change “prognosis” to “prognostic”. Line 72, add “of” between “especially” and “poor neurodevelopmental outcomes.”

Ans: Thank you for your suggestion. I have corrected these grammatical errors.

METHODS:

1. RE: Can the authors comment on what proportion of eligible infants were included in the study, and if any were excluded what the reasons for exclusion were?

Ans: We added a new figure (Figure 1) to illustrate the inclusion /exclusion criteria and the participants flow through the trial. The description was also added in Section 2. Materials and Methods (Page 7, Line 72-89).

2. RE: The authors should describe how they chose the two-point approach to measuring growth velocity from among the alternative methods available (eg Patel et al, Calculating Postnatal Growth Velocity in Very Low Birth Weight (VLBW) Premature Infants, J Perinatal 2009).

Ans: This is an important question. There was no standardization of methods used to calculate preterm infant growth velocity. According to a systematic review (Fenton TR, et al. Preterm Infant Growth Velocity Calculations: A Systematic Review. Pediatrics. 2017), the most frequently used method reported in the 1543 studies was g/kg/d (40%), followed by g/d (32%); 29% reported change in z score. The majority of studies that reported g/kg/d calculations did not report what they used for the weight as denominator (n = 94 [62%]). Of the studies that reported the denominators, the majority used an average of the start and end weights as the denominator (36%) followed by exponential methods (23%), and birth weight (10%). So we chose the most common method (two-point approach) and the z-score change to calculate growth velocity. Thank you for the literaure you provided (Patel et al, Calculating Postnatal Growth Velocity in Very Low Birth Weight (VLBW) Premature Infants, J Perinatal 2009). It seems that exponential method is superior (more accurate). We will consider use this method to evaluate the GV in the future. We have rewritten the section 2.4 outcome measurement for further explaination and cited these references.

Otherwise, we also used the change in weight z-score within 1 week to evaluate the weight gain velocity as follows (figure c and d). But the correlation between weight z core change and protein intake is weak, probably because the difference in z score was subtle within a short period (one week). So the results was not shown in figure 1.

3. RE: Please describe how extreme values were handled. In my experience, there are often a few extreme values in any relatively large dataset of milk samples analyzed by MIRIS (often due to the sample not being fully homogenized). Were there any extreme values and if so were they included or discarded? If discarded, what criteria were used to determine which samples were excluded?

Ans: We agree with your experience. These extreme values were included and analyzed in our study. We hoped we could reduce the effect of extreme values by collecting more samples and calculating the weekly average (every weekly data consists of 14 milk samples). 

4. RE: Describe what is meant by the covariate "bottle feeding" - is that bottle feeding on the day the sample was taken? Bottle feeding ever? Similarly, describe what is meant by "ventilator support" - on the day of the sample? Ever?

Ans: We observed that bottle feeding may consume more energy than feeding via a nasogastric tube and may affect infant’s growth in the clinical practice. Otherwise, ventilator support may reduce the breathing work and the energy consumption of infants. So, we added these 2 cofactors in the model. Both “bottle feeding” and “ventilator support” meant that was used in some days of the week when the sample was taken. We added the above explanation below table 4 and table 5 (Line 224-226, 233-235)

5. RE: The authors state in the discussion that they could not use length or head circumference (HC) measurements due to measurement error. It may be worth adding to the methods section that the intention was to assess length and HC, and how they determined that the measurements were too inaccurate to use.

Ans: We have added this to method section (Line 143-145) and explanation about inaccuracy in discussion section (Line 293-299) as follows” Firstly, we measured the head circumference (HC) and body length weekly, and calculated growth velocity as centimeters/week (cm/wk). However, some of the growth velocity (cm/wk) was minus. The possible cause is that the time interval is too short to show the subtle change in HC and length. Due to inaccurate measurement, we didn’t analyze the relationship between HC, length growth and daily protein intake.” 

RESULTS:

1. RE: In Table 1: The title of the right column is "weekly sample" - so are these results describing individual infants, or total samples (eg were 60% of infants in the study male or 60% of samples came from male infants?). Also clarify what is meant by "Post menstrual age;" is this the PMA at the time the sample was taken? For duration of TPN support, I suspect the variable is quite skewed; consider reporting median rather than mean and standard deviation.

Ans: I’m sorry about the unclear description. The results of table 1 are description total weekly sample (n=245). For example, 61.2% of weekly data came from male infants. We also added explanation (*) for PMA and z-score below table 1 as follow: PMA and z-score represented the infant age and weight status at time of growth measurements. According to your suggestion, we have reported the variable of TPN duration as median value in table 1 and added the description in section 2.5. statistical analysis (Page 11, Line 162-164)

2. RE: Can the authors comment on the proportion of samples that were donor milk versus maternal milk? There are well described differences between the macronutrient content of donor vs maternal milk, as the authors have described in the discussion.

Ans: The proportion of samples that were donor milk (12.7%) was added in Table 1. It means total or part of milk samples were donor milk within 1 week. 

3. RE: Line 52, remove "Nevertheless"

Ans: Thank you for your suggestion. We have removed it.

DISCUSSION: Overall the organization of the discussion could be improved, and the authors could more clearly state how their findings add to what is already known on this topic. Also, while the remainder of the paper has excellent English language, the discussion has a few minor English errors that should be corrected; I have attempted to note them here to assist the authors in clarifying their language.

1. RE: The first paragraph is very long. Consider breaking it up into several paragraphs and clearly identify the key point of each paragraph.

Ans: We have broken it up to three paragraph and compressed the section of discussion.

2. RE: Line 187, add that the linear association was positive. Line 190 clarify what "it" refers to. Line 191, consider changing "amount" to "content of milk" or "intake" depending on the authors' intention

Ans: Thank you for your comment. We have reworded that as above. (Page 20, Line 238, 241-242) 

3. RE: Lines 211-212: clarify what "higher" means (higher than what?)

Ans: Thank you for your suggestion. We had compressed the section of discussion and deleted this paragraph.

4. RE: Line 222, consider describing the protein content of the HMF rather than the number of packs

Ans: We have added the protein content of the HMF (1.0-1.1g/100ml) in Line 274.

5. RE: Line 227, clarify if the authors are referring to the current study or to previous studies. Lines 227-235; this paragraph would benefit from better organization to determine the authors' key point

Ans: We removed this paragraph due to little correlation with our study.

6. RE: Line 237, I think the authors mean "protein intake" rather than "fortification" here

Ans: We have rewritten this sentence (Line 277-278). 

7. RE: Line 257, by "was not provided in our results" do the authors mean it was not found in their analysis or was not studied?

Ans: We meant that it was not studied (analyzed). We have deleted this sentence due to vague description.

8. RE: Correct the English language: Line 189, remove "the" before growth. Line 198, "observation" should be "observational." Lines 201-202 I would suggest "Clinical practice and nutrition guidelines have changed substantially over the past two decades." Line 207 change "the human milk sample" to "human milk samples." Lines 211-212, consider changing "approximate" to "support," and add "the" before "intrauterine fetus." Line 217, add "with" before "body weight." Line 221, change "while" to "when." Line 240, "systemic" should be "systematic." Line 251, "target" should be "targeted." Line 254, "consist" should be "exist." Line 260, "group" should be "groups." Lines 263-264 need to be rewritten and clarified. Line 277, "prematurity" should be "premature infants"

Ans: Thank you for these comments. We have corrected these language errors. The whole manuscript was edited for language and writing (section Acknowledgement, Line 319-320)

9. Can the authors comment on generalizability of their results based on the inclusion criteria of their study?

Ans: We have added the description about generalizability as follows: “Finally, we only enrolled in-hospital premature infants with a gestational age of 25 to 36 weeks who reached full enteral feeding and under exclusive human milk diet. Whether the protein content in human milk also plays a crucial role in the premature infants under mixed feeding with formula milk, with parental nutrition support, and the infants after hospital discharge, remains to be determined.” (Line 302-307)

Again, thank you for giving us the opportunity to strengthen our manuscript with your valuable comments and queries. We have worked hard to incorporate your feedback and hope that these revisions persuade you to accept our submission.

Sincerely,

Yi-Hsuan Lin, MD

Corresponding Author

Ming-Chih Lin, M.D. Ph.D. 

Division of Neonatology & Pediatric Cardiology

Department of Pediatrics,

Taichung Veterans General Hospital,

Taichung city, Taiwan

TEL: 886-4-23592525-5901

FAX: 886-4-23741359

---

## [Decision Letter · Decision Letter 1]

6 Dec 2019

PONE-D-19-17439R1

The impact of macronutrients in human milk on the growth of preterm infants

PLOS ONE

Dear Dr. Lin,

Thank you for submitting your manuscript to PLOS ONE. After careful consideration, we feel that it has merit but does not fully meet PLOS ONE’s publication criteria as it currently stands. Therefore, we invite you to submit a revised version of the manuscript that addresses the points raised during the review process. Please carefully review and address remaining concerns raised by reviewer and remove causal language.

We would appreciate receiving your revised manuscript by Jan 20 2020 11:59PM. To enhance the reproducibility of your results, we recommend that if applicable you deposit your laboratory protocols in protocols.io, where a protocol can be assigned its own identifier (DOI) such that it can be cited independently in the future. For instructions see: http://journals.plos.org/plosone/s/submission-guidelines#loc-laboratory-protocols

We look forward to receiving your revised manuscript.

Kind regards,

Melissa F. Young, Ph.D.

Academic Editor

PLOS ONE

Reviewers' comments:

Reviewer's Responses to Questions

**Comments to the Author**

1. If the authors have adequately addressed your comments raised in a previous round of review and you feel that this manuscript is now acceptable for publication, you may indicate that here to bypass the “Comments to the Author” section, enter your conflict of interest statement in the “Confidential to Editor” section, and submit your "Accept" recommendation.

Reviewer #2: (No Response)

Reviewer #3: (No Response)

2. Is the manuscript technically sound, and do the data support the conclusions?

Reviewer #2: (No Response)

Reviewer #3: Partly

3. Has the statistical analysis been performed appropriately and rigorously? 

Reviewer #2: (No Response)

Reviewer #3: No

4. Have the authors made all data underlying the findings in their manuscript fully available?

Reviewer #2: (No Response)

Reviewer #3: Yes

5. Is the manuscript presented in an intelligible fashion and written in standard English?

Reviewer #2: (No Response)

Reviewer #3: Yes

6. Review Comments to the Author

Reviewer #2: The authors have done a good job responding to the reviewers' comments. The discussion is still long but is OK as is.

Reviewer #3: The authors have addressed the majority of my concerns. The length of the paper and clarity of the figures and English language in the manuscript are much improved. However, a few key items remain:

MAJOR ISSUES:

STATISTICAL ANALYSIS

- All analyses need to account for correlated data. The analysis of protein intake and weight gain (Table 5) still states it is linear regression rather than generalized estimating equations.

- The authors report measuring changes in weight Z-score (p 10, line 138) but I do not see any reported results using weight Z-scores. Either this line should be removed or the authors should report the Z-score results.

- The authors should describe why they chose the covariates included in the models (eg a prior based on known associations with infant growth and milk macronutrient content, or what criteria they used if the covariates were chose in some other fashion).

DATA DESCRIPTION

- The authors still need to provide more clarity in the manuscript regarding the distinction between individual infants who contributed to the study vs milk samples included the study. Demographic data for the infants themselves should be reported somewhere, even if just in a supplementary file, rather than just the sample-level data shown in Table 1. The figure they show in response to my first review comment (detailing the number of data points each infant contributed) should be included in the manuscript or at least in some form of supplementary material.

ADDITIONAL COMMENTS BY SECTION:

OVERALL:

- The language still suggests causal relationships rather than association in a few places, for example the title (I suggest "association of macronutrients in human milk with growth..." rather than "impact of macronutrients on growth") and Conclusion (I suggest "positive association with" rather than "effect on" body weight gain), and see line 199 in Results.

- The authors still use the unclear term "catch-up growth" in several places (eg line 241, line 260, line 319).

ABSTRACT:

-In Methods, "full enteral feeding" suggests no additional nutritional support so authors can remove "without parenteral nutrition support" which is redundant and creates confusion. Add a description of what the outcome variable was and how it was measured.

- What do the authors mean by "highly" associated (line 48)?

INTRODUCTION:

- I disagree with the statement that breastmilk is protective against postnatal growth failure (line 64); rather, formula-fed infants often grow faster than milk-fed infants, but breastmilk is preferred despite that because of its other protective effects (eg on NEC, neurodevelopment, obesity prevention).

METHODS:

- Clarify what criteria were used to determine if milk samples were "insufficient" (line 87)?

- Line 97 remove "(96.5%)" which is redundant (and remove "up to" in line 96 if the number is known).

- Line 135 clarify if the intake includes protein intake from fortifier too.

- Line 151, I appreciate the description of bottle feeding and ventilator use in the table footnotes, but it should be included here in the text in the description of covariates.

- Line 158, I believe the authors mean "32 to <37" weeks

- Line 164, clarify which variables are exposure vs outcome variables

RESULTS:

- Line 197-8, clarify the meaning of the effect size, eg if this is describing the results of Table 4, "After adjustment, each 1 g/kg/day of protein content in milk was associated with 3.6 g/kg/day greater weight gain." Where did the number 5.70 come from? Same comment for lines 200-201 regarding protein intake. Results should only be described one time so if the authors discuss the results of Tables 4 & 5 later, it doesn't also need to be included here.

Line 198, the authors state only protein was associated with weight gain, but the table shows an association with energy too.

Lines 212-214: This description of choosing GEE belongs in the Methods section.

DISCUSSION:

- Line 170, do the authors mean "achieved" rather than "required" here?

TABLES

Table 1: Is PMA normally distributed? If not, may be useful to show median/interquartile range instead. Also, clarify what weight z-score means; was it at the start of the week or end or an average?

Table 2: Title could be clarified to "Macronutrient concentration in human milk samples (N=245)". I think it would improve clarity to move the PMA at time of sample from Table 1 to Table 2, if the authors agree.

FIGURES

- Figures 1 & 2. Good additions to the paper.

- Figure 3. Does this show adjusted or unadjusted values?

7. PLOS authors have the option to publish the peer review history of their article (what does this mean?). If published, this will include your full peer review and any attached files.

Reviewer #2: Yes: Ekhard Ziegler

Reviewer #3: No

---

## [Author Response · Author response to Decision Letter 1]

19 Dec 2019

Dear Dr. Young: 

Thank you for inviting us to submit a revised draft of our manuscript entitled, “Association of macronutrients in human milk with the growth of preterm infants” to PLOS ONE again. We also appreciate the time and effort you and each of the reviewers have dedicated to providing insightful feedback on ways to strengthen our paper. Thus, it is with great pleasure that we resubmit our article for further consideration. We have incorporated changes that reflect the detailed suggestions you have graciously provided. We also hope that our edits and the responses we provide below satisfactorily address all the issues and concerns you and the reviewers have noted.

To facilitate your review of our revisions, the following is a point-by-point response to the questions and comments.

To Reviewer #3: 

Thank you for your comments and suggestions. The questions are answered below: 

MAJOR ISSUES:

STATISTICAL ANALYSIS

1. RE: All analyses need to account for correlated data. The analysis of protein intake and weight gain (Table 5) still states it is linear regression rather than generalized estimating equations.

Ans: I’m sorry about the wrong title of table 5. The result of table 5 is from a generalized estimating equation. I have corrected the title of table 5 (Page 19, Line 243)

2. RE: The authors report measuring changes in weight Z-score (p 10, line 138) but I do not see any reported results using weight Z-scores. Either this line should be removed, or the authors should report the Z-score results.

Ans: The correlation between weight z core change and protein intake is weak, probably because the difference in z score was subtle within a short period (one week). So the results was not reported. I have removed the description in section method (p 10, line 141-142). 

3. RE: The authors should describe why they chose the covariates included in the models (eg a prior based on known associations with infant growth and milk macronutrient content, or what criteria they used if the covariates were chose in some other fashion).

Ans: We added the description as follows, “Univariate analysis for possible confounding factors was performed. All the factors significant (p<0.05) on univariate analysis are entered in the multivariate analysis model. In addition, PDA, SGA and gestational age at birth were known important cofactors for weight growth of preterm infants [26, 27]. We also included these three factors in the multivariate analysis model, even not statistically significant in univariate analysis.” (Page 12, Line 176-181)

DATA DESCRIPTION

4. RE: The authors still need to provide more clarity in the manuscript regarding the distinction between individual infants who contributed to the study vs milk samples included the study. Demographic data for the infants themselves should be reported somewhere, even if just in a supplementary file, rather than just the sample-level data shown in Table 1. The figure they show in response to my first review comment (detailing the number of data points each infant contributed) should be included in the manuscript or at least in some form of supplementary material.

Ans: We have clarified that some infants contributed more than 1 weekly sample if their hospital stay was longer (Page 12, Line 181-182). Demographic data for included 99 infants was displayed in S3 Table and was cited in the manuscript (Page 14, Line 195-196). The number of data points each infant contributed was also reported in S2 Figure (supporting information) and was cited in the manuscript (Page 8, Line 110). 

ADDITIONAL COMMENTS BY SECTION:

OVERALL:

5. RE: The language still suggests causal relationships rather than association in a few places, for example the title (I suggest "association of macronutrients in human milk with growth..." rather than "impact of macronutrients on growth") and Conclusion (I suggest "positive association with" rather than "effect on" body weight gain), and see line 199 in Results.

Ans: We have corrected the words in title, results, and conclusion (Line 2, Line 214, Line 332).

6. RE: The authors still use the unclear term "catch-up growth" in several places (eg line 241, line 260, line 319).

Ans: We have deleted the term in line 256, line 274, line 332.

ABSTRACT:

7. RE: In Methods, "full enteral feeding" suggests no additional nutritional support so authors can remove "without parenteral nutrition support" which is redundant and creates confusion. Add a description of what the outcome variable was and how it was measured.

Ans: We have deleted the redundant words (Line 38) and added the description about outcome variable (Line 41-42).

8. RE: What do the authors mean by "highly" associated (line 48)?

Ans: We have deleted the word “highly” in line 50.

INTRODUCTION:

9. RE: I disagree with the statement that breastmilk is protective against postnatal growth failure (line 64); rather, formula-fed infants often grow faster than milk-fed infants, but breastmilk is preferred despite that because of its other protective effects (eg on NEC, neurodevelopment, obesity prevention).

Ans: Thank you for your comment. According to the reference 1 (Section on B. Breastfeeding and the use of human milk. Pediatrics. 2012;129(3):e827-41), lower rate of long-term growth failure is due to lower NEC incidence. We agree with you that most studies revealed that breast-fed infants are leaner than formula-fed infants. We have deleted this part to avoid confusion (Line 65).

METHODS:

10. RE: Clarify what criteria were used to determine if milk samples were "insufficient" (line 87)?

Ans: Each weekly average data contains 14 human milk samples as shown in Figure 2. We defined insufficient human milk sample as more than one missing sample within a week. We have added description about that (Line 88-89)

11. RE: Line 97 remove "(96.5%)" which is redundant (and remove "up to" in line 96 if the number is known).

Ans: We have removed these redundant words (Line 100).

12. RE: Line 135 clarify if the intake includes protein intake from fortifier too.

Ans: Yes, the energy and protein amount from fortifier were included in the calculation of total protein intake (g/kg/d) and protein-to-energy (PE) ratio (g/100 kcal). We have clarified it in Line 137-138. 

13. RE: Line 151, I appreciate the description of bottle feeding and ventilator use in the table footnotes, but it should be included here in the text in the description of covariates.

Ans: We have added the definition of bottle feeding and ventilator use in section 2.4 Cofactors (Line 158-162) as follows, “The cofactor of bottle feeding means oral feeding without an orogastric tube applied on some days of the week when samples were taken. Ventilator support was defined as noninvasive positive pressure ventilators or nasal continuous positive airway pressure was used on some days of the week when the sample was taken.”

14. RE: Line 158, I believe the authors mean "32 to <37" weeks

Ans: Yes. We have corrected this mistake. (Line 166)

15. RE: Line 164, clarify which variables are exposure vs outcome variables

Ans: We have modified the sentence as follows: “The main exposure variables (protein concentration in human milk and total protein intake amount) and outcome variable (body weight gain velocity) were all normally distributed.” (Line 172-174)

RESULTS:

16. RE: Line 197-8, clarify the meaning of the effect size, eg if this is describing the results of Table 4, "After adjustment, each 1 g/kg/day of protein content in milk was associated with 3.6 g/kg/day greater weight gain." Where did the number 5.70 come from? Same comment for lines 200-201 regarding protein intake. Results should only be described one time so if the authors discuss the results of Tables 4 & 5 later, it doesn't also need to be included here.

Ans: For effect size estimation, under alpha level of 0.05 and 90% power, the sample size of this study can detect the primary outcomes as minimal as 5.70 g/100ml (protein concentration) and 2.28 g/kg/day (daily protein intake). It is describing the result of Figure 3 (without adjustment with models). I’m sorry about unclear description. To avoid confusion, we have removed this part (Line 212-213, 216).

17. RE: Line 198, the authors state only protein was associated with weight gain, but the table shows an association with energy too.

Ans: We have deleted the statement “There were no other significant associations of human milk components with growth” (Line 213-214).

18. RE: Lines 212-214: This description of choosing GEE belongs in the Methods section.

Ans: We have deleted the description in results section (Line 227-229).

DISCUSSION:

19. RE: Line 170, do the authors mean "achieved" rather than "required" here?

Ans: Yes. I have corrected the word (Line 284).

TABLES

20. RE: Table 1: Is PMA normally distributed? If not, may be useful to show median/interquartile range instead. Also, clarify what weight z-score means; was it at the start of the week or end or an average?

Ans: the Kolmogorov–Smirnov test was used, and PMA is not normally distributed. It was reported as median (interquartile range) in table 1. We also clarified that z-score means the weight status in the first day of week (Line 203-204).

21. RE: Table 2: Title could be clarified to "Macronutrient concentration in human milk samples (N=245)". I think it would improve clarity to move the PMA at time of sample from Table 1 to Table 2, if the authors agree.

Ans: Thank you for your suggestions. We have revised the title of table 2 according to your suggestion (Line 208). However, the PMA means the infant age at the time of growth measurement rather than the lactation stage. The human milk may be collected earlier by mothers and stored in refrigerator at −20°C. It means that the infant may be fed with colostrum (stored in refrigerator) when he was 1 month old. Otherwise, the results of table 2 contains some data of donor milk, and the lactation stage (PMA) is uncertain. So we are sorry about that we couldn’t clarify the PMA (lactation stage) in table 2.

FIGURES

22. RE: Figures 1 & 2. Good additions to the paper.

Ans: Thank you for your comment.

23. RE: Figure 3. Does this show adjusted or unadjusted values?

Ans: Figure 3 showed unadjusted values. We have added this explanation in figure legend. (Page 18, Line 223)

 

Again, thank you for giving us the opportunity to strengthen our manuscript with your valuable comments and queries. We have worked hard to incorporate your feedback and hope that these revisions persuade you to accept our submission.

Sincerely,

Yi-Hsuan Lin, MD

Corresponding Author

Ming-Chih Lin, M.D. Ph.D. 

Division of Neonatology & Pediatric Cardiology

Department of Pediatrics,

Taichung Veterans General Hospital,

Taichung city, Taiwan

TEL: 886-4-23592525-5901

FAX: 886-4-23741359

---

## [Decision Letter · Decision Letter 2]

24 Feb 2020

PONE-D-19-17439R2

The association of macronutrients in human milk with the growth of preterm infants

PLOS ONE

Dear Dr. Lin,

Thank you for submitting your manuscript to PLOS ONE. After careful consideration, we feel that it has merit but does not fully meet PLOS ONE’s publication criteria as it currently stands. Therefore, we invite you to submit a revised version of the manuscript that addresses the points raised during the review process. Please carefully review and address reviewer feedback. Manuscript requires editing for grammar and journal formatting requirements.

We would appreciate receiving your revised manuscript by Apr 09 2020 11:59PM. To enhance the reproducibility of your results, we recommend that if applicable you deposit your laboratory protocols in protocols.io, where a protocol can be assigned its own identifier (DOI) such that it can be cited independently in the future. For instructions see: http://journals.plos.org/plosone/s/submission-guidelines#loc-laboratory-protocols

We look forward to receiving your revised manuscript.

Kind regards,

Melissa F. Young, Ph.D.

Academic Editor

PLOS ONE

Reviewers' comments:

Reviewer's Responses to Questions

**Comments to the Author**

1. If the authors have adequately addressed your comments raised in a previous round of review and you feel that this manuscript is now acceptable for publication, you may indicate that here to bypass the “Comments to the Author” section, enter your conflict of interest statement in the “Confidential to Editor” section, and submit your "Accept" recommendation.

Reviewer #2: All comments have been addressed

Reviewer #3: All comments have been addressed

2. Is the manuscript technically sound, and do the data support the conclusions?

Reviewer #2: Yes

Reviewer #3: Yes

3. Has the statistical analysis been performed appropriately and rigorously? 

Reviewer #2: Yes

Reviewer #3: Yes

4. Have the authors made all data underlying the findings in their manuscript fully available?

Reviewer #2: Yes

Reviewer #3: Yes

5. Is the manuscript presented in an intelligible fashion and written in standard English?

Reviewer #2: Yes

Reviewer #3: Yes

6. Review Comments to the Author

Reviewer #2: The authors have done a decent job of improving the manuscript.

It stands to hope that the findings of the study lead to improved nutritional care of premature infants in Taiwan.

A few minor points:

Line 35: Change "enrolled" to "is based on data from"

Line 38: Change "from" to "of"

Line 53: Change "have" to "had"

Line 68: Change "higher" to "high"

Line 69: Change "might not" to "are

Line 157: Does "noninvasive" mean "non-intubated"?

Line 140-142: Should read: "Weight growth velocity was obtained by subtracting the first from the last weight of the week divided by 7"

Reviewer #3: The authors have addressed my previous comments. There are a few very minor errors (noted below) that can easily be corrected during the copyediting process for publication.

1. Interquartile ranges should be presented as: median (25%ile, 75%ile) or the authors can present the range as median (min, max); but not usually median (#) as currently shown because it is unclear what the # represents

2. Check the % of singletons in Supplementary Table 3; it doesn't match the N given

3. There are a few very minor grammatical errors so the authors should proofread the manuscript for attention to English language.

4. The discussion is somewhat repetitive and could be edited for better organization and clarity, but it does not contain any errors and could be published as is.

7. PLOS authors have the option to publish the peer review history of their article (what does this mean?). If published, this will include your full peer review and any attached files.

Reviewer #2: Yes: Ekhard Ziegler

Reviewer #3: No

---

## [Author Response · Author response to Decision Letter 2]

26 Feb 2020

Dear Dr. Young: 

Thank you for inviting us to submit a revised draft of our manuscript entitled, “Association of macronutrients in human milk with the growth of preterm infants” to PLOS ONE again. We also appreciate the time and effort you and each of the reviewers have dedicated to providing insightful feedback on ways to strengthen our paper. Thus, it is with great pleasure that we resubmit our article for further consideration. We have incorporated changes that reflect the detailed suggestions you have graciously provided. We also hope that our edits and the responses we provide below satisfactorily address all the issues and concerns you and the reviewers have noted.

To facilitate your review of our revisions, the following is a point-by-point response to the questions and comments.

To Reviewer #2: 

Thank you for your comments and suggestions. The questions are answered below: 

1. RE: A few minor points:

Line 35: Change "enrolled" to "is based on data from"

Line 38: Change "from" to "of"

Line 53: Change "have" to "had"

Line 68: Change "higher" to "high"

Line 69: Change "might not" to "are

Ans: Thank you for these comments. We have corrected these words.

2. Line 157: Does "noninvasive" mean "non-intubated"?

Ans: Yes. Noninvasive positive pressure ventilators (NIPPV) mean non-intubated ventilator support from nasal prong. It also called nasal intermittent positive pressure ventilators (NIPPV). We have replaced the term with nasal intermittent positive pressure ventilators (Line 157). 

3. Line 140-142: Should read: "Weight growth velocity was obtained by subtracting the first from the last weight of the week divided by 7"

Ans: We are sorry about unclear description. We have revised this sentence (Line 141-142). The equation of weight growth velocity was displayed in Fig 2.

To Reviewer #3: 

Thank you for your comments and suggestions. The questions are answered below: 

1. RE: Interquartile ranges should be presented as: median (25%ile, 75%ile) or the authors can present the range as median (min, max); but not usually median (#) as currently shown because it is unclear what the # represents

Ans: We have revised it. The interquartile ranges are reported as median (25%ile, 75%ile) (Table 1). 

2. RE: Check the % of singletons in Supplementary Table 3; it doesn't match the N given

Ans: We are sorry about this error. We have corrected it (Supplementary Table 3). 67.7% of enrolled infants were singletons. 

3. RE: There are a few very minor grammatical errors so the authors should proofread the manuscript for attention to English language.

Ans: We have corrected some grammatical errors. The whole manuscript was edited for English language and writing (section Acknowledgement, Line 330-331)

4. RE: The discussion is somewhat repetitive and could be edited for better organization and clarity, but it does not contain any errors and could be published as is.

Ans: Thank you for your suggestion. 

Again, thank you for giving us the opportunity to strengthen our manuscript with your valuable comments and queries. We have worked hard to incorporate your feedback and hope that these revisions persuade you to accept our submission.

Sincerely,

Yi-Hsuan Lin, MD

Corresponding Author

Ming-Chih Lin, M.D. Ph.D. 

Division of Neonatology & Pediatric Cardiology

Department of Pediatrics,

Taichung Veterans General Hospital,

Taichung city, Taiwan

TEL: 886-4-23592525-5901

FAX: 886-4-23741359

---

## [Editor Report · Decision Letter 3]

10 Mar 2020

The association of macronutrients in human milk with the growth of preterm infants

PONE-D-19-17439R3

Dear Dr. Lin,

We are pleased to inform you that your manuscript has been judged scientifically suitable for publication and will be formally accepted for publication once it complies with all outstanding technical requirements.

With kind regards,

Melissa F. Young, Ph.D.

Academic Editor

PLOS ONE
---

## [Editor Report · Acceptance letter]

13 Mar 2020

PONE-D-19-17439R3 

The association of macronutrients in human milk with the growth of preterm infants 

Dear Dr. Lin:

I am pleased to inform you that your manuscript has been deemed suitable for publication in PLOS ONE. Congratulations! Your manuscript is now with our production department. 

With kind regards,

on behalf of

Dr. Melissa F. Young 

Academic Editor

PLOS ONE